

# A cross-sectional study on interference control: age affects reactive control but not proactive control

Yanfang Peng[1,2], Qin Zhu[3], Biye Wang[4] and Jie Ren[5]

[1] School of Physical Education and Sport Training, Shanghai University of Sport, Shanghai, Shanghai, China
[2] School of Sport Science, Wenzhou Medical University, Wenzhou, Zhejiang, China
[3] Division of Kinesiology and Health, University of Wyoming, Laramie, WY, USA
[4] School of Sport Science, Yangzhou University, Yangzhou, Jiangsu, China
[5] China Table Tennis College, Shanghai University of Sport, Shanghai, Shanghai, China

## ABSTRACT

**Background:** Working memory updating (WMU), a controlled process to continuously adapt to the changing task demand and environment, is crucial for cognitive executive function. Although previous studies have shown that the elderly were more susceptible to cognitive interference than the youngsters, the picture of age-related deterioration of WMU is incomplete due to lack of study on people at their middle ages. Thus, the present study investigated the impact of age on the WMU among adults by a cross-sectional design to verify whether inefficiency interference control accounts for the aging of WMU.

**Methods:** In total, 112 healthy adults were recruited for this study; 28 old adults (21 female) ranging from 60 to 78 years of age; 28 middle-age adults (25 female) ranging from 45 to 59 years of age; 28 adults (11 female) ranging from 26 to 44 years of age; and 28 young adults (26 female) ranging from 18 to 25 years of age. Each participant completed a 1-back task. The inverse efficiency score was calculated in various sequences of three trials in a row to quantify the performance of WMU for adults of various ages.

**Results:** Inverse efficiency score of both young groups (young adult and adult) were significantly shorter than the old group in both Repeat-Alternate (RA, including □□○ and ○○□) and Alternate-Alternate (AA, including ○□○ and □○□) sequential patterns and they were additionally better than the middle-age group in AA sequential pattern.

**Conclusion:** With the increase of difficulty in the task, the difference in reactive interference control between young and middle age was gradually revealed, while the difference between young and old remained to apparent. The degradation of WMU aging may begin from middle-age and presents selective impairment in that only reactive interference control, but not proactive interference control, shows pronounced age-related decline. The preliminary results can inform future studies to further explore the whole lifespan trajectories of cognitive functions.

Corresponding author
Jie Ren, renjie@sus.edu.cn

## INTRODUCTION

Cognitive aging has been a central theme (*Anderson & Craik, 2017*) of the growing global aging because the compromised cognitive functioning in later life is a risk factor for increased morbidity and mortality (*Bruce et al., 1995*). Effective cognitive function can help old adults to maintain independence and promote quality of life in old age (*Royall et al., 2005*). The purpose of exploring age-related cognitive changes is to discover an aging mechanism and to provide preventative interventions. In the past two decades, a substantial body of research has documented the decline of cognition for old adults compared to young adults. The progression of aging is unclear because middle age has been understudied, however, investigating this period of the lifespan is important for the understanding of senescence.

Since WM is capacity-limited, working memory updating (WMU) is used to continuously adapt to changing task demands and environment (*Morris & Jones, 1990*), and is crucial for cognitive executive function. Updating is a process of dismantling and recreating associations between content and context (*Artuso & Palladino, 2018*). That is, unbinding the outdated contents from their contexts in time and establishing new bindings (*Oberauer, 2009*). It is hard to distinguish between outdated and new items particularly when they're similar. Many studies have proved that the WMU of young adults over older adults was better, because older adults decreased the ability to modulate brain activation and they could not maintain a better availability of attention representations to efficiently inhibit irrelevant information (*Arjona, Escudero & Gómez, 2016*; *Fiore et al., 2011*; *Podell et al., 2012*; *Sambataro et al., 2015*). Old adults need larger accuracy cost and an additional focus-switch cost, which is the cost of switching attention to relevant information (*Schmiedek, Li & Lindenberger, 2009*; *Verhaeghen & Basak, 2005*), especially old-old adults (*Borella et al., 2007*; *De Beni & Palladino, 2004*; *Kato et al., 2016*). The inefficient inhibition of irrelevant information was thought to be the crucial reason for WMU aging. But *Borella, Carretti & De Beni (2008)* recruited 304 subjects age-range 20–86 to measure four kinds of WM tasks and two kinds of inhibition tasks. Results showed that inhibition was not as a crucial contributor to age-related decline in the functional capacity of WM across the adult life-span as previously thought. Recently, *Dagry, Vergauwe & Barrouillet (2017)* suggested that during WMU, attention was allocated to capture current goals but not to inhibit stimulus that should be ignored.

Control processes are a critical component of the WM function (*Braver, Gray & Burgess, 2012*). When the task contains high interference, the task will tax more cognitive control resources to protect the contents of memory against interference (*Szmalec et al., 2011*). Control processes in WMU are responsible for selecting relevant information; preventing interference; updating at appropriate junctures and so on (*Braver, Gray & Burgess, 2012*). *Kessler & Meiran (2006, 2008)* pointed out there are two dissociable independent components contributing to WMU, one is used to modify the relevant representations in memory, the other one is used to protect the contents of WM against interference. Some empirical studies also embraced this view (*Artuso & Palladino, 2011*; *Rac-Lubashevsky & Kessler, 2016b*). For example, in the N-back task, the mismatch

trials containing a switch cost was performed slower than the match trials, and the lure trials (no-longer/no-yet relevant item corresponds to the currently presented item) were performed slower than the mismatch trials due to more similarity-based interference they caused (*Szmalec et al., 2011*). The memory of previous trials affected the current performance (*Rac-Lubashevsky & Kessler, 2016b*; *Szmalec et al., 2009*). How does the control process work? The Dual Mechanisms of Control theory stated that two types of cognitive control are dynamically interacting with each other. One is reactive control, which is a transient stimulus-driven. It is responsible for "just-in-time" selection by detecting and solving interference after it occurs. The other one is proactive control, which is anticipatory goal-driven. It is responsible for early selection by anticipating and preventing interference before it occurs (*Braver, 2012*; *Braver, Gray & Burgess, 2012*). Which cognitive control is the crucial contributor to WMU aging? Some evidence showed that age selectively impaired cognitive control. Recent studies reported that old adults had selective difficulty in memorizing content–context associations but not in isolated contents (*Artuso et al., 2017*; *Artuso & Palladino, 2011*; *McCormick-Huhn et al., 2018*; *Old & Naveh-Benjamin, 2008*; *Pelegrina et al., 2012*) and the delays in selection were longer with a function of memory load (*Artuso et al., 2017*), implying that proactive control was impaired with aging. But *Xiang et al. (2016)* reported that older adults had selective deficits in reactive control.

Previous literature comparing young with old adults found an aging effect on WMU, however, the progression of age-related changes and the crucial element causing WMU aging were unknown because of the exclusion of middle-age adults in the previous studies. Previous studies suggested that besides vocabulary knowledge, which increases with aging (*Miller & Lachman, 2003*; *Salthouse, 2010*; *Singh-Manoux et al., 2012*), most cognitive functions decline with aging and present variant recession cycles. For example, content–context binding (*Cowan et al., 2006*; *Hommel, Kray & Lindenberger, 2011*; *Siegel, 1994*; *Swanson, 2017*) and switch (*Kray, Eber & Lindenberger, 2004*; *Reimers & Maylor, 2005*) decrease approximately linearly across the adult life-span. The decline of memory, process speed, inhibition and reasoning begin from middle age (*Anstey et al., 2015*; *Borella, Carretti & De Beni, 2008*; *Davis et al., 2017*; *Hughes et al., 2018*; *Persad et al., 2002*; *Singh-Manoux et al., 2012*; *Zimprich & Mascherek, 2010*). Although the timing of age-related decline of cognitive functions varies, middle age looks like an important age boundary.

This study focused on verifying whether inefficient interference control is a crucial contributor to WMU aging across the adult life span and employed 1-back test, which includes two stimuli. The interference difficulty was manipulated by adjusting the sequence relationship. Sequential interference had an accumulative effect in N-back test (*Oberauer et al., 2013*; *Salmi et al., 2018*; *Soetens, Boer & Hueting, 1985*) and only lure trial N−1 showed a significant sequence effect in 1-back test (*Rac-Lubashevsky & Kessler, 2016b*), which had been proved by previous studies. Four sequence patterns were distinguished by the sequence of three trials in a row. The main objective is to assess age-changes of WMU in four sequence patterns. If WMU aging is attributed to inefficient interferent control, the age-related decline was steeper as interference increases.

**Table 1  Participant demographics.**

| Age group | N | Mean age | BMI | Females | MMSE score |
|---|---|---|---|---|---|
| Young-adults | 28 | 22.71 (1.67) | 20.4 (1.8) | 26 | – |
| Adults | 28 | 31.68 (4.63) | 22.8 (2.8) | 11 | – |
| Middle-aged adults | 28 | 54.21 (4.06) | 23 (2.7) | 25 | – |
| Old | 28 | 67.68 (4.37) | 23 (2.2) | 21 | 27.8 (2.2) |

**Note:**
SD in brackets. MMSE, Mini-Mental State Examination.

The hypotheses stated that (1) WMU performance will decline with interference increase, (2) the WMU degradation begins from middle-age and becomes steeper in old age and (3) age-related decline is getting steeper as interference increases.

# METHODS AND MATERIALS

## Ethical approval

This study received approval from the Ethics Committee of Shanghai University of Sport (No. 2017044).

## Participants

The sample consisted of 112 adult participants, 28 older adults (21 female) ranging from 60 to 78 years of age ($M = 67.68$, SD $= 4.23$); 28 middle-age adults (25 female) ranging from 45 to 59 years of age ($M = 54.25$, SD $= 4.84$); 28 adults (11 female) ranging from 26 to 44 years of age ($M = 31.68$, SD $= 4.63$); and 28 young adults (26 female) ranging from 18 to 25 years of age ($M = 22.71$, SD $= 1.67$), recruited through Shanghai University of Sport and local communities. Participants were compensated for a small gift for their participation. All participants didn't have cognitive impairment as tested by the Mini-mental State Examination and signed the informed consent (see Table 1).

## Stimuli

The 1-back task was adopted from *Rac-Lubashevsky & Kessler (2016a, 2016b)* to observe a participant's ability to update information. This task involved the continuous presentation of a solid gray square "□" (side length 38 mm) or a solid gray circle "○" (the diameter of 38 mm). A Lenovo computer with a 17-inch VGA display (frequency 60 Hz, resolution 1,366 × 768) was used for stimulus presentation, and the Matlab2015 software package (Psychtoolbox 3.0) was used for response sampling. All stimuli were presented on a white background.

## Task

Participants were instructed to monitor stimuli subtended a visual angle of 5.73° horizontally and 5.73° vertically from viewing distance 55 cm and decided whether the presented shape was the same as the one that had been presented immediately before by pressing one of two keys on a response box: "3" for "yes" and "1" for "no". The test consisted of a practice block and two formal experimental blocks. The practice block contained 20 trials with feedback to familiarize the participant with the task. The system

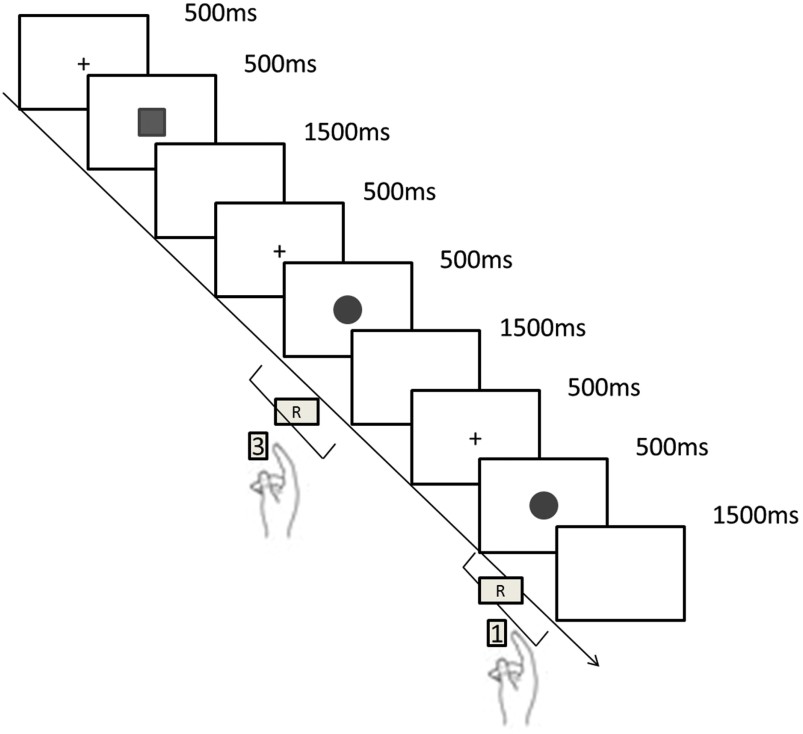

**Figure 1 The 1-back task procedure.** The task started with a blank fixation screen presented for 500 ms , the stimulus "□" or "○" was then presented for 500 ms randomly, followed by a white display for 1,500 ms. Participants were required to respond quickly and accurately as soon as the target stimuli appeared. If it was the same as the previous trial pressing "1" (same), if not pressing "3" (difference). The maximum duration for response to be made was 2,000 ms (the presentation of the stimulus 500 ms + the presentation of white display 1,500 ms).

automatically transitions to the formal experimental session until the accuracy reached up to 66.7% in the practice session, otherwise, the system remained in the practice session. The data of the practice session was not recorded.

Each block of the formal test included 42 trials and started with a blank fixation screen presented for 500 ms, the stimulus "□" or "○" was then presented for 500 ms randomly, followed by a white display for 1,500 ms. Participants were required to respond quickly and accurately as soon as the target stimuli appeared. The maximum duration for a response to be made was 2,000 ms (the presentation of the stimulus 500 ms + the presentation of white display 1,500 ms). Reaction time and accuracy were automatically recorded by Matlab2015. If the participant didn't respond in time (RT > 2,000 ms), the reaction time was recorded as an error. Participants were assured with adequate rest between two blocks of the experiment (Fig. 1).

## Statistical analysis

Previous studies reported that only the lure trial (trial N−1) influenced the current performance (Trial N+1) in the 1-back task (*Rac-Lubashevsky & Kessler, 2016b*, *2016a*; *Soetens, Boer & Hueting, 1985*; *Szmalec et al., 2011*). Accordingly, the data were classified by the sequence of three trials in a row, and four sequence patterns could be

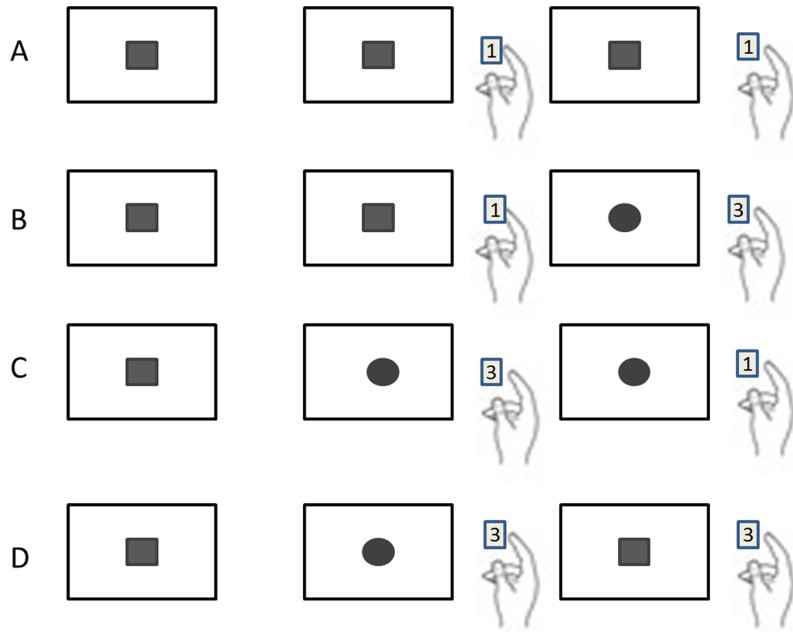

**Figure 2 Four different sequential patterns with the sequence of three trials in a row.** (A) Repeating the same stimulus three times is named as RR. (B) Repeating the same stimulus twice followed by a different stimulus (named as RA; including ○○□ and □□○). (C) Altering the stimulus in first two followed by repeating the second stimulus on the third (named as AR; including ○□□ and □○○). (D) Altering the stimuli twice in the three trials (named as AA; including ○□○ and □○□).

identified: (1) Repeating the same stimulus three times (named as RR; including □□□ and ○○○); (2) Repeating the same stimulus twice followed by a different stimulus, (named as RA; including ○○□ and □□○); (3) Altering the stimulus in the first two followed by repeating the second stimulus on the third (named as AR; including ○□□ and □○○); (4) Altering the stimuli twice in the three trials (named as AA; including ○□○ and □○□) (Fig. 2).

RT data were cleaned by removing inaccurate and no response trials. RT below 100 ms was treated as response error, and outlying trials which were more than three standard deviations above each sequence patter condition mean were removed. Time cost and accuracy are known to be negatively related (*Pachella, 1974*). Particularly, with increasing difficulty, participants may decide to emphasize either speed or accuracy. Disregard accuracy or analyzing the accuracy and RTs separately impair the power to detect relationships and interactions (*Hughes et al., 2014*). Consequently, inverse efficiency score (IES), RT with consideration of response accuracy, has been proposed as a good way (*Bruyer & Brysbaert, 2011*; *Townsend & Ashby, 1978*) and be used to evaluate updating in this study. IES is calculated as RT divided by PC (the proportion of correct responses). The formula is IES = RT/PC. Performance and IES are negatively related, a lower IES corresponds to better performance. A 4 × 4 (Sequence pattern × Age) mixed-design ANOVA with sequence pattern (RR, RA, AR and AA) as within-subjects factor and Age

**Table 2 Age group comparison between Young adult (18–25 age), Adult (26–44 age), Middle-age (45–59 age), Old age (60 age) about four sequential patterns.**

| Sequential pattern | Age groups (ms) | | | | Simple main effect of Age (F) | Post hoc Tukey HSD test |
|---|---|---|---|---|---|---|
| | 18–25 ① | 26–44 ② | 45–59 ③ | 60 ④ | | |
| RR | 538.2 (24.2) | 608.9 (37.4) | 685 (37.3) | 843.2 (73.2) | 1.5 | |
| RA | 686.2 (30.1) | 716.7 (46.4) | 1,004.6 (56.4) | 1,110 (64.2) | 3.87** | ① < ④* ② < ④* |
| AR | 807.9 (42.3) | 965.6 (77.5) | 1,030 (48.2) | 1,210.2 (90.6) | 2.42 | |
| AA | 1,340.8 (93.2) | 1,406.5 (106.9) | 2,488.1 (256.6) | 2,472.7 (243.8) | 35.79** | ① < ③** ② < ③** ① < ④** ② < ④** |

**Notes:**
$^{*}$ $p < 0.05$.
$^{**}$ $p < 0.01$

(18–25 age, 26–44 age, 45–59 age and ≥60 age) as a between-subjects factor was employed to examine the effects of interference and age on WMU.

## RESULTS

The two-way ANOVA showed significant main effects for Age ($F$ (3, 108) = 17.43, $p < 0.001$, $\eta_p^2 = 0.326$), Pattern ($F$ (3, 108) = 132.27, $p < 0.001$, $\eta_p^2 = 0.551$). Tukey post-hoc comparisons showed that young adults outperformed middle-age ($p < 0.001$) and old ($p < 0.001$), the same results appeared between adults and middle-age ($p < 0.01$) and old ($p < 0.001$); the difference were not observed both between young adults and adults group ($p = 0.725$) and middle-age and old group ($p = 0.554$). That implied the serious degradation of WMU began in middle-age. The significant differences among the four sequence patterns were also observed, implying the declined performance with increasing interference, IES of RR was the lowest, then RA, and then AR, AA was the highest ($p < 0.001$).

The interaction between Age and Pattern ($F$ (3, 324) = 6.55, $p < 0.001$, $\eta_p^2 = 0.121$) reached significance. The simple main effect analysis showed a significant age effect in RA ($F$ (3, 432) = 3.87, $p < 0.01$) and AA ($F$ (3, 432) = 35.79, $p < 0.001$), the post hoc Tukey's HSD test showed that both young groups outperformed the old group in RA, and they outperformed both middle-age and old groups in AA (see Table 2). In summary, WMU declined with increasing age, however, the selective age-related impaired was presented (see Fig. 3).

## DISCUSSION

The study aimed to analyze the impact of age on WMU interference control by a cross-sectional comparison. Updating requires holding temporary binding between contents and contexts and unbinding outdated contents in time because WM capacity is limited. Cognitive control should hold maximal flexibility to find an optimal balance between maintaining and replacing to ensure performance, especially on a strong

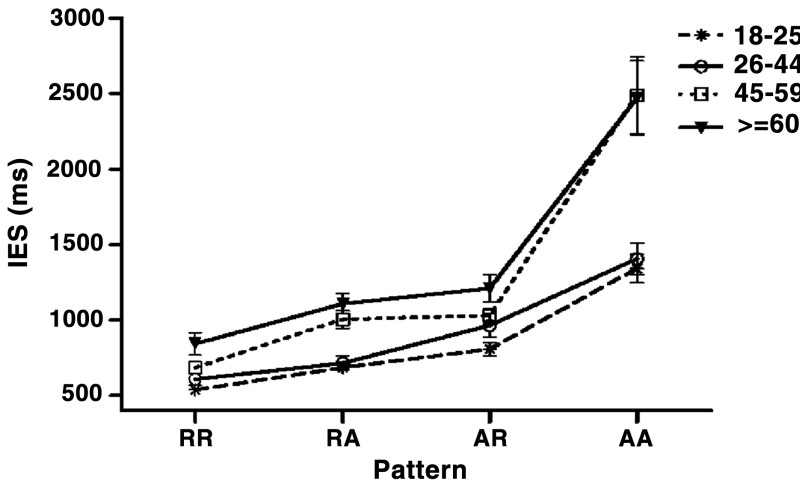

**Figure 3** **The age-related change during various sequence patterns.** *X* axis is sequence pattern, *Y* axis is the inverse efficiency score (RTc/PC). Error bars represent standard error.

interference task. In this study, the interference difficulty was manipulated by adjusting the sequence relationship, four sequence patterns including RR, RA, AR and AA patterns were classified by three trials in a row. The participants were divided into the youngster, middle-aged and the elderly. Besides, the young participants were subdivided into young adults and adults, because middle-aged had been understudied in previous aging studies, potentially due to that the difference between middle age and other groups was not observed (*Phillips et al., 2011*). Subdivision of youth groups could help us to acquire a better understanding of the progression of age-related changes across adult life-span particularly from young to middle age. Our results showed two young groups outperformed the old group in both RA and AA and they were additionally better than the middle-age group in AA sequential pattern.

Four sequential patterns were classified by three trials in a row. As we expected, the performance declined with interference increase. *Rac-Lubashevsky & Kessler (2016a, 2016b)* separated several contributions to updating by reference-back task. The reference-back task contains reference trials, which is presented inside a red frame, and comparison trials, which is presented inside a blue frame. The participants are required to judge whether the presented stimulus is the same as, or different from, the previous reference trials. In other words, the comparison trials should be compared to the previous reference trials, but should not be updated. Compared with comparison trials, reference trials contain an additional updating cost. In this study, repetition trials as comparison trials should not be updated, the cost was smaller than alternation trials (RR < RA; AR < AA). The previous study of two-alternative forced-choice task or 1-back choice task found AR elicited a stimulus–response (S–R) conflict response which contains additional time cost and accurate cost than other patterns (*Rac-Lubashevsky & Kessler, 2016b*; *Szmalec et al., 2009*). But the response of 1-back depended on the relationship between the previous trial and the current stimulus. AA requires the participant to switch focus to relevant

information and to prevent similarly interference in terms of ensuring a currently appropriate action (*Wylie & Allport, 2000*). In this study, an accumulative effect of the sequence interference was observed, especially in AA (RR < AR; RA < AA), suggesting two independent sources of contribution to updating. That was consistent with previous studies (*Artuso & Palladino, 2011*; *Kessler & Meiran, 2006*, *2008*; *Rac-Lubashevsky & Kessler, 2016b*).

A better WMU of young adults over older adults was proved in this study as well as in many previous studies (*Artuso et al., 2017*; *Guerreiro, Murphy & Van Gerven, 2013*; *Hommel, Kray & Lindenberger, 2011*; *Pelegrina et al., 2012*; *Phillips et al., 2011*; *Schmiedek, Li & Lindenberger, 2009*). And the superior performance of young groups to the middle age group was observed in this study. No difference was found between young adult and adult. No difference was found between middle-aged and old adults on total mean IES. Only two stimuli were employed in this study to induce larger similarity-based interference, used IES to increase the detection power, and shortened ISI to prevent participators from refreshing during free time. Even under the circumstances, the difference between young and middle age was observed only in the pattern with the biggest interference (AA). The task of previous studies was possibly too easy to observe the early onset of cognitive impairment at middle age. This is possibly the reason why middle-aged understudied in previous aging studies.

Age-related changes varied among four sequential patterns and only appeared in RA and AA. The difference between young and old adults was observed in RA and AA. RA and AA contain a characteristic that requires updating the new item and replacing the outdated one after the onset of new items. The age-related decline was observed in alternation trials disregarding prior sequence characteristics, suggesting that age only affected just-in-time selection. The reactive control, but not proactive control, declined with aging (*Xiang et al., 2016*). That challenged earlier views of age can only affect proactive control (*Botvinick et al., 2001*; *Braver & West, 2015*). The difference between young and middle-aged was observed in AA, but not in RA. Compared with RA, AA elicits more similarity-based interference and requires more cognitive control than RA. The difference between young and middle-aged was gradually revealed with the increase of interference, while the difference between young and old remained to be apparent, reflecting the degradation of WMU begins from the middle-age. The age-related decline only appeared in mismatch trials, implying age-related switch deficit might be a crucial contributor to WMU aging across the adult lifespan. However, previous switch aging studies suggested that special switch cost, which represented the differences between switch and nonswitch trials within a block as in this study, were largely unrelated to age; only the global switch, the ability to efficiently coordinate multiple tasks, was negatively affected by age (*Kray & Lindenberger, 2000*; *Mayr & Liebscher, 2001*). Using varied tasks may cause different results. The N-back task may be a measure of cognitive control when it involves higher interference (*Szmalec et al., 2011*). And its interference can continuously increase by manipulating the sequence relationship. The task-induced huge similarity-based interference due to employing only two stimuli in this study and the IES

index was adopted to ensure the power. All of this may prompt some new findings. Age-related decline in reactive interference control but not in proactive control was gradually revealed after middle age and the decline got steeper with age. In the future, research, particularly including middle age, should consider the impact of task difficulty and the power of indexing on the result. And future studies should explore further the relationship between updating, interference, and aging using the N-back task.

## Limitations

The present study had a few limitations. Firstly, the main motivation of the study is to explore the progression of WMU aging by including the middle age group, but the sample size of the cross-sectional design was too small to further explore timing detail of onset of aging and the progression of aging after 60 age. Secondly, the cross-sectional design could not avoid individual differences within age groups. A larger sample of participants should be recruited in future studies and combine with cross-sectional and longitudinal follow-up design, which may offset the defect of the present study. Finally, the lack of information on education and other socioeconomic variables of the studied population may limit the generalization of the findings.

## CONCLUSIONS

In conclusion, with the increase of difficulty in the task, the difference in reactive cognitive control between young and middle age was gradually revealed, while the difference between young and old remained to be apparent. The results reflected that WMU degradation may begin from the middle age and become steeper in old age. WMU aging presents selective impairment. Only reactive interference control, but not proactive interference control, shows pronounced age-related decline, which mainly reflects a larger special switch cost. Age-related switch decline may be a crucial contributor to WMU in aging. The preliminary results can inform future studies to further explore the whole lifespan trajectories of cognitive functions.

### Funding

This work was supported by the Soft Science Research Project of Wenzhou Science and Technology Bureau (No. R20170042) The funders had no role in study design, data collection and analysis, decision to publish, or preparation of the manuscript.

### Grant Disclosures

The following grant information was disclosed by the authors:
Soft Science Research Project of Wenzhou Science and Technology Bureau: R20170042.

### Competing Interests

The authors declare that they have no competing interests.

## Author Contributions

- Yanfang Peng conceived and designed the experiments, performed the experiments, analyzed the data, prepared figures and/or tables, authored or reviewed drafts of the paper, and approved the final draft.
- Qin Zhu analyzed the data, authored or reviewed drafts of the paper, and approved the final draft.
- Biye Wang performed the experiments, prepared figures and/or tables, and approved the final draft.
- Jie Ren conceived and designed the experiments, prepared figures and/or tables, and approved the final draft.

## Human Ethics

The following information was supplied relating to ethical approvals (i.e., approving body and any reference numbers):

The Ethics Committee of Shanghai University of Sport granted ethical approval to carry out the study within its facilities (Ethical Application Ref: 2017044).

## Data Availability

The raw measurements are available as a Supplemental File.

## Supplemental Information

Supplemental information for this article can be found online at http://dx.doi.org/10.7717/peerj.8365#supplemental-information.

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
