# Peer review of "A cross-sectional study on interference control: age affects reactive control but not proactive control"

_PeerJ, doi:10.7717/peerj.8365_

## Round 0.1 · original submission · Major Revisions

To summarize, Reviewer 1 has a number of important questions and comments for you to address in your revision, point by point. Reviewer 2 is skeptical of cross-sectional aging studies with small samples, preferring the strength of longitudinal studies. Since that is not possible here, my suggestion is to scale back your conclusions (e.g., as preliminary results that can inform future studies), and to note the limitations of cross-sectional studies in a "Limitations" section. Reviewer 2 also notes the absence of information on education and other socioeconomic variables for each age group (Table 1 does not break down education by age group). If more specific demographic information is not available, this should also be noted as a limitation.

Reviewer 1 ·

Basic reporting

In the current work the authors administered a WM updating task (modeled after the well-known n-back task) to adults and older adults and distinguished the two components of proactive interference control and reactive interference control. The authors found that only reactive interference control is impaired through aging.
I found the ms interesting and the topic addressed timely and not yet widely investigated. Hovewer I have some issues that could be addressed to improve the ms.


The focus of the ms -that is the difference and theoretical importance of proactice/reactive interference control in updating- is not adequately addressed. The authors should anticipate explanation of these two constructs through introduction and not only define them in the discussion section (see lines 235-239). In addition, they should discuss more thoroughly the work already cited on updating, and add further work that could help to enlarge the focus of the relationship between updating, interference and aging, such as Artuso & Palladino, 2016 (CP); Hartman et al., 2001 (ANC); Oberauer & Vockenberg, 2009, (QJEP).

Please double check English throughout all the ms by a native English speaker.

Experimental design

As the authors state that the MMSE was administered, they should report mean scores and sds for the groups of older

How the authors decided the age ranges, and labels? I found the labels sometimes odds and confusing, such as old-young adults or young-young adults. Consider for example Belacchi & Artuso (2018) P&A, for more convincing labels, e.g. young adults, adults, middle-aged adults, young old and old old.
In addition, I wonder whether it is possible to have two groups of older, i.e., young old (60-75) and old old (> 75) in order to observe possible differences, e.g. a more pronounced impairment in the old old or no differences.

In Table 1, change the label “Values” to “Mean age values”

Please modify some typos, e.g., line 70 “Carretti” instead of “Grretti” or line 168 “below” instead of “blow” and many others.

Validity of the findings

Conclusions are well stated and robust.

Reviewer 2 ·

Basic reporting

No comment

Experimental design

This is a study on a relevant topic using generally sound experimental methods. The pattern of results fits the suggested idea that reactive interference starts to decline earlier during the lifespan than the switch costs associated with updating contents of working memory. This main motivation of the study – of exploring age-related changes in between the age ranges typically investigated in age-comparative experimental cognitive aging studies – is challenged by severe limitations of the study design, however. Just as much as I agree that charting the whole lifespan trajectories of cognitive functions is important, I am also convinced that this can only be achieved with longitudinal cohort-sequential studies (or approximated using really large and representative cross-sectional samples, if one is willing and able to make assumptions about cohort differences). The kind of small-sample age-group comparison employed in the present study has its value in identifying certain effects that generally differentiate younger from older adults – to be then further investigated with more elaborate study designs. Trying to get at non-linear trajectories across adulthood or identifying points during the lifespan when certain declines begin or accelerate, this approach is clearly limited, particularly if selection effects and differential representativeness of the different samples cannot be ruled out. Given the small size of the four age groups and the lack of information on education and other socio-economic variables, the validity of conclusions like “WMU degradation begins in middle age and becomes steeper in old age” is not ensured. As the young-middle-old age comparison is the main contribution of the paper, I regret that I cannot support its acceptance.

Validity of the findings

see above

---

## Round 0.2 · Minor Revisions

Thank you for your revised manuscript, which is much improved. However, Reviewer 1 noted a number of typos that have not been corrected yet. I would also suggest additional editing for English language. Another recommendation is to define RA and AA in the abstract: Repeat-Alternate (RA) and Alternate-Alternate (AA).

Reviewer 1 ·

Basic reporting

The manuscript is original and sufficient background is provided. However, there are still many editing typos that were already raised in the previous revision and were not addressed.

Experimental design

Clear, and original design

Validity of the findings

Novel and original

Additional comments

The authors adequately addressed all the points raised. I have no further comments.
However, there are many typos/editing issues (see APA style). Some of them are listed below; all the manuscript should be carefully checked for other similar typos that denote inaccurate writing.

line 55: “new bindings(K Oberauer, 2009)” should be “new bindings (Oberauer, 2009)”

line 65: “Borella and Garretti (2008)” should be “Borella and Carretti (2008)”

line 79: “(Y Kessler & Meiran, 2008; Yoav Kessler & Meiran, 2006)” should be “ (Kessler & Meiran, 2006; 2008)

line 80: “ (Rac-Lubashevsky & Kessler, 2016b)(Artuso & Palladino, 2011)” should be “(Artuso & Palladino, 2011; Rac-Lubashevsky & Kessler, 2016b)”

line 221: “(Phillips, Bull et al. 2011).” should be written in the same font of the manuscript

Artuso & Palladino (2018) reference is missing

Borella & Carretti (2008) reference is missing

---

## Round 0.3 · accepted · Accept

Thank you for your careful revisions to the manuscript.